# Positive Linear Relationship between Nucleophosmin Protein Expression and the Viral Load in HPV-Associated Oropharyngeal Squamous Cell Carcinoma: A Possible Tool for Stratification of Patients

**DOI:** 10.3390/ijms24043482

**Published:** 2023-02-09

**Authors:** Marco D’Agostino, Marco Di Cecco, Carla Marani, Maurizio Giovanni Vigili, Sara Sileno, Chiara Costanza Volpi, Annunziata Gloghini, Daniele Avitabile, Alessandra Magenta, Siavash Rahimi

**Affiliations:** 1Experimental Immunology Laboratory, Istituto Dermopatico dell’Immacolata, IDI-IRCCS, 00167 Rome, Italy; 2Division of Histopathology, Ospedale San Carlo di Nancy, 00165 Rome, Italy; 3Head and Neck Surgery Departments, Istituto Dermopatico dell’Immacolata, IDI-IRCCS, 00167 Rome, Italy; 4Institute of Translational Pharmacology IFT, National Research Council of Italy (CNR), 00133 Rome, Italy; 5Diagnostic Pathology and Laboratory Medicine Department, Fondazione IRCCS Istituto Nazionale Tumori, 20133 Milan, Italy; 6Idi Farmaceutici S.R.L., Pomezia, 00071 Rome, Italy; 7Anatomic Pathology Department, Istituto Dermopatico dell’Immacolata IDI-IRCCS, 00167 Rome, Italy

**Keywords:** oropharyngeal squamous cell carcinoma, HPV, nucleophosmin, inflammation

## Abstract

Most oropharyngeal squamous cell carcinomas (OPSCCs) are human papillomavirus (HPV)-associated, high-risk (HR) cancers that show a better response to chemoradiotherapy and are associated with improved survival. Nucleophosmin (NPM, also called NPM1/B23) is a nucleolar phosphoprotein that plays different roles within the cell, such as ribosomal synthesis, cell cycle regulation, DNA damage repair and centrosome duplication. NPM is also known as an activator of inflammatory pathways. An increase in NPM expression has been observed in vitro in E6/E7 overexpressing cells and is involved in HPV assembly. In this retrospective study, we investigated the relationship between the immunohistochemical (IHC) expression of NPM and HR-HPV viral load, assayed by RNAScope in situ hybridization (ISH), in ten patients with histologically confirmed p16-positive OPSCC. Our findings show that there is a positive correlation between NPM expression and HR-HPV mRNA (Rs = 0.70, *p* = 0.03), and a linear regression (r^2^ = 0.55; *p* = 0.01). These data support the hypothesis that NPM IHC, together with HPV RNAScope, could be used as a predictor of transcriptionally active HPV presence and tumor progression, which is useful for therapy decisions. This study includes a small cohort of patients and, cannot report conclusive findings. Further studies with large series of patients are needed to support our hypothesis.

## 1. Introduction

Oral squamous cell carcinoma (OSCC) and oropharyngeal squamous cell carcinoma (OPSCC) are the most common types of HNSCC [1,2,3]. It is well known that tobacco smoking and alcohol consumption are the main causes of HNSCC [4,5], and despite the reduction in these risk factors, the incidence of HNSCC has increased [6]. Most OPSCCs are high-risk (HR) human papillomavirus (HPV)-related [7,8,9], with HPV16, the most common genotype involved, representing 86.7% of all HPV-positive OSCCs, followed by HPV18 and HPV33 [10].

Patients with HPV-induced OPSCC have been shown to be younger and are less likely to have a history of tobacco or alcohol use than patients with HPV-negative OPSCCs. A superior socioeconomic status and sexual behaviors have also been associated with the development of HPV-positive OPSCC [11]. Patients with HPV-associated OPSCC showed a better clinical outcome compared to those with HPV-negative tumors, with a good response to chemoradiotherapy. Hence, HPV status is included as a stratification factor for clinical trials using patients with OPSCC [12,13].

The HPV genome consists of a small double-stranded DNA and includes three major regions. The early genes (E1-E7) are expressed early in the viral infectious cycle for the regulation of transcription, plasmid replication and transformation. They encode the E6 and E7 oncoproteins, which are responsible for tumorigenesis [6]. The late genes code for the major (L1) and minor (L2) capsid proteins involved in the packaging of the viral genome and virus release. The long control region (LCR) contains the regulatory elements for transcription and replication. E6 protein binds the tumor suppressor p53 that is involved in apoptosis and cellular senescence, promoting its degradation [14]. Moreover, the E7 protein interacts and inactivates the retinoblastoma (Rb) tumor suppressor, which plays an important role in the regulation of the cell cycle. The inactivation of p53 and pRb causes uncontrolled cell proliferation and apoptosis inhibition, resulting in cellular transformation and genomic instability [15]. E6 and E7 expression inhibition in oropharyngeal cancer cells is associated with the restoration of p53 and Rb pathways and increased apoptosis [16].

The nucleolus is known to have a central role in the interaction between the virus and the infected cell. Indeed, several viruses, including HPV, have evolved molecular strategies to take control of nucleolar functions, ultimately favoring viral replication and maturation, such as ribosomal biogenesis, cell cycle progression and apoptosis [17].

The E7 protein has been colocalized with pRB in the nucleolus [18]. It interacts with p14arf and the Upstream Binding Factor 1 (UBF1), a key factor in the activation of RNA polymerase I machinery, thus favoring rDNA transcription [19]. The expression and/or localization of other nucleolar proteins, such as PICT-1 [20] and nucleolin [21], has been shown to correlate with HPV18-dependent carcinogenesis [21].

In keeping with the nucleolar role of HPV infection, changes in numbers or in mor-phometric parameters of silver-stained nucleolar organizer regions (AgNORs) have been previously proposed as useful predictors of worse prognosis in different forms of HPV- infected squamous cell carcinomas [22,23,24,25]. Recently, the presence of hyperchromasia, thickening of nuclear contour, and prominent nucleoli on Pap smear have been proposed as strong indicators of the presence of intraepithelial neoplasia grade 1 (CIN1) or greater CIN 2-3 [26]. Indeed, AgNOR areas greater than 3.3 µm^2^ with concomitant expression of p16INK4a have been shown to be useful in the identification of high-risk human papillomaviruses (HR-HPV) for the development of cervical cancer [27].

Nucleophosmin (NPM, also called NPM1/B23) is a nucleolar phosphoprotein that plays different roles within the cell, such as ribosomal synthesis, cell cycle regulation, repair of DNA damage and centrosome duplication [28].

NPM interacts with different proteins from RNA and DNA viruses, including HIV, HCV, HBV and HPV, and affects the infection process [29]. In fact, NPM is implicated in different steps of the viral replicative cycle, including cytoplasmic nuclear traffic and the final assembly of viral particles, thus impacting the viral replication efficiency [29]. It has been shown that NPM levels increased in human foreskin keratinocytes (HFK) overexpressing E7, following differentiation induced by methylcellulose. Furthermore, in cultured cells and organotypic structures overexpressing E7 and E6/E7, an NPM protein level upregulation was found [30]. In addition, NPM downregulation in E6/E7 overexpressing cells provoked induction of p53 and pRb levels, causing replicative capacity reduction and an increase in the differentiation marker, Keratin 1, in HFKs [30].

Therefore, NPM is required for proliferation and inhibition of differentiation in primary keratinocyte cells expressing HPV’s E6 and E7 [30]. The host nucleolar protein NPM is also required during HPV16 pseudovirus (PsV) assembly since it interacts with L2 minor capsid protein, playing an important role in establishing the correct interactions between L2 and L1 during the assembly of the HPV capsid [31].

Given the importance of the nucleolus in HPV infection and the pivotal role of the nucleolar protein NPM in the replicative capacity of HPV-infected cells, the aim of this study is to elucidate the relationship between NPM and the HR-HPV viral load, assessed by RNAScope, in p16-positive HPV-associated OPSCC.

## 2. Results

All cases show non-keratinized (“basaloid”) and mixed-type (“keratinized” and “basaloid”) SCC morphology [32] and positive p16 immunohistochemical staining according to the Lewis et al., criteria (Figure 1b) [33].

RNAScope-ISH revealed positive staining for E6/E7 mRNA HR-HPV (Figure 2a,c). There was a different degree of intensity of staining among the cases. In some cases, the strong signals formed clusters, indicating a high viral load. The staining was specific and limited only to the neoplastic cells and dysplastic epithelium. The lymphoid stroma and normal squamous epithelium were negative.

In a similar way, we found that NPM IHC positivity was also variable in FFPE tissue. Some cases had low IHC positive staining, whereas others showed intense areas of positive staining (Figure 2b,d).

Interestingly, NPM IHC intensity positively correlated with high-risk HPV mRNA-IHS in OPSCC (Rs = 0.70, *p* = 0.03) (Figure 3). Moreover, a linear relationship was found between NPM IHC and HR-HPV mRNA, detected by RNAScope (Figure 3). Our data indicate that NPM is a positively signed direct regressor for HR-HPV mRNA expression. Thus, NPM expression level could be a predictor of HR-HPV mRNA levels with an average increase of 1.15 units for each 1-unit increase in NPM expression level. The “true” mean increase could plausibly be expected to lie between 0.30 and 2.00 (95% confidence interval). The r^2^ suggests that about 55% of the variability in the HR-HPV mRNA can be explained by its relationship with NPM expression. The results show that NPM and E6/E7 mRNA HR-HPV positivity were very similar in patients’ categorization.

## 3. Discussion

The majority of the OPSCCs are HR-HPV-dependent and present a good clinical outcome in terms of chemoradiotherapy sensitivity compared with HPV-negative HNSCC [34]. p16 is a tumor suppressor protein and belongs to the family of INK4 cyclin-dependent-kinase inhibitors. Immunohistochemistry with p16 is an excellent surrogate of high-risk HPV infection [35,36].

However, approximately 20% of p16-positive OPSCC patients may lack transcriptionally-active HR-HPV [37,38,39,40].

In addition, a positive immunohistochemical reaction with p16 shows the presence or absence of HR-HPV infection and does not assess the amount of the viral load. Assuming that the amount of HPV viral load does establish the progression of the disease, then the p16 immunohistochemical test could not define the aggressiveness of the disease. RNAScope complementary to HPV E6/E7 mRNA is a molecular technique, which allows visualizing viral transcripts in FFPE tissue. HPV-RNAScope demonstrates high sensitivity and specificity for detection of HPV infection and has been proposed as the clinical standard for assigning a diagnosis of HPV-positive OPSCC [41,42,43,44].

Viral proteins often target the nucleolus, nucleolar proteins are released into the nucleoplasm and cytoplasm, and in most cases, viral nucleic acids are localized in the nucleolus. On the other hand, cellular proteins can be relocated to the nucleolus and participate in the infectious process [17]. Nucleolar proteins, in association with viral proteins, participate in replicating viruses in the capsid assembly and are often implicated in the transport of viral particles in the nucleoplasm and cytoplasm. Indeed, several viruses, including HPV, take advantage of nucleolar functions in order to increase viral replication and maturation, controlling ribosomal biogenesis, cell cycle progression and apoptosis [17].

The expression of different biomarkers has been linked to HPV carcinogenesis. Given the importance of the nucleolus, different nucleolar proteins have been used to determine the progression of HPV infection. For example, PITC1 [20] and nucleolin [21] are two nucleolar proteins whose expression has been linked to uterine cervical cancer aggressivity. Additionally, the expression of another cellular protein, proliferating cell nuclear antigen (PCNA), a protein essential for DNA replication, has been used for this aim [24]. PCNA expression was associated with high-risk HPV and the progression of cervical intraepithelial neoplasia (CIN).

In keeping with the nucleolar role of HPV infection, the numbers and the morphometric parameters of AgNORs have been proposed as useful predictors of worse prognosis in different forms of HPV-infected squamous cell carcinomas [22,23,24,25].

Further, hyperchromasia, thickening of nuclear contour, and prominent nucleoli on Pap smear have been used as indicators of intraepithelial neoplasia grade 1 (CIN1) or greater CIN 2–3 [26]. Moreover, AgNOR areas and p16INK4a positivity have been associated with HR-HPV for the development of cervical cancer [27].

Since NPM is a nucleolar protein that is frequently overexpressed in different types of tumours [45], we investigated the possible correlation between NPM and HR-HPV viral load in a small cohort of HPV-related OPSCC, using immunohistochemistry and RNAScope, respectively.

Our results showed a positive correlation and a linear relationship between the expression of NPM and HPV E6/E7 mRNA. These data support the hypothesis that NPM could be used as a predictor of transcriptionally active HPV presence and possibly tumor progression.

NPM up-regulation is required for the inhibition of differentiation and promotion of proliferation in differentiating cells, which express E6–E7 [30]. In addition, NPM has a role in establishing the correct interactions between L2 and L1 during HPV capsid assembly [31].

We previously showed that NPM intracellular protein level is increased by inflammatory cytokines and is released into the extracellular space by fibroblasts and primary keratinocytes upon cytokine stimulation associated with psoriasis [46].

Interestingly, circulating NPM is upregulated in the plasma of psoriatic patients and positively correlated with the severity of the disease and some determinants of cardiovascular risk [46].

Moreover, we demonstrated that NPM acts as an alarmin after genotoxic stress treatment and, when it is secreted in the extracellular space, induces inflammation by binding the Toll-Like Receptor 4 (TLR4) receptor. This interaction activates a signaling pathway that leads to the translocation of NF-kB in the nucleus with consequent induction of inflammatory genes, such as interleukin 6 (IL-6) and cyclooxygenase 2 (COX-2) [47].

We know that an increased level of reactive oxygen and nitrogen species (ROS and RNS), produced during chronic inflammation, play a pivotal role in tumor progression, and it is established that NPM is a modulator of the cellular oxidative stress response and an activator of inflammation pathway. Given the fact that inflammation-mediated DNA damage provides a potential mechanism of HPV integration and that an excessive amount of ROS and RNS promote DNA damage [48], it would be reasonable to speculate that NPM could potentially contribute to HPV integration and the promotion of tumor progression.

Targeting a single host protein to simultaneously impact different steps of viral replication is an appealing solution. Since NPM induction increases viral replication and also cell cycle progression, targeting NPM might simultaneously impact on viral replication and on the neoplastic state that emerges from it [29].

Different groups are currently studying the development and preclinical/clinical evaluation of NPM inhibitors. Among these, only two have been evaluated in Phase I/II cancer clinical trials, the CIGB-300 peptide in HPV-positive cervical cancers [49] and the synthetic pseudopeptide N6L (NCT01711398) in all solid tumors. CIGB-300 has also been tested in an HIV infection model with success [50].

Moreover, in cervical cancers treatment, 75% of the patients showed tumor reduction at colposcopy and 19% exhibited full histological regression. In addition, HPV DNA was negative in 48% of the previously positive patients. Long-term follow-up did not show adverse events or recurrences [49].

Since we demonstrated in previous papers that NPM could be considered an extra-cellular alarmin, the circulating expression of NPM could also be exploited in future applications as a predictor of HPV viral load by salivary or blood test screening in OPSCC.

In conclusion, our results show that a combination of tests, immunohistochemistry of NPM and HR-HPV RNAScope-ISH could be used in routine practice to stratify patients with HPV-positive OPSCC into a group with relatively stable disease and a group with the progression of malignancy. This finding potentially could have clinical implications because NPM inhibition could offer therapeutic benefits to these patients.

This study includes a small cohort of patients and, by definition, cannot be regarded as yielding conclusive findings. Further studies with a larger cohort of patients are needed to support this hypothesis.

## 4. Materials and Methods

### 4.1. Patients

Ten cases of OPSCC were retrieved from archives of Histopathology Division of San Carlo di Nancy Hospital and Istituto Dermopatico dell’Immacolata (IDI-IRCCS), Rome-Italy. Haematoxylin and eosin (H&E) stained slides obtained from formalin-fixed paraffin-embedded (FFPE) lesion tissue were examined.

### 4.2. Immunohistochemistry (IHC)

Leica Stainer Immunohistochemistry System was used for the p16 antibody (BD Biosciences, purified mouse anti-human p16, product number 550834, concentration 31.25 µg/mL, dilution 1:10). Appropriate positive and negative control tissue was used.

NPM IHC was performed on FFPE tissue as previously described [46]. Briefly, sections were incubated with the anti-human NPM rabbit polyclonal antibody (ab15440; dilution 1:200). Rabbit IgG isotype control (Santa Cruz Biotechnology, Dallas, TX, USA) was used at the same concentration of primary antibody. Polyclonal biotinylated secondary Abs and staining kits were obtained from Vector Laboratories. Immunoreactivity was visualized using the peroxidase reaction with 3-amino-9-ethylcarbazole (AEC, Vector Laboratories, Burlingame, CA, USA) in H_2_O_2_ as a substrate and samples were counterstained with hematoxylin. As a negative control, primary Abs were omitted. Stained sections were analyzed with the AxioCam digital camera coupled to the Axioplan 2 microscope (Carl Zeiss AG, Oberkochen, Germany). NPM staining intensity was evaluated by quantitative analysis (Image J color deconvolution) in three adjacent fields of each section by two independent observers, not aware of the status of the specimens. The mean values for each patient’s section expressed as arbitrary units (A.U.) of NPM IHC positivity area were used for correlation and regression analyses.

### 4.3. RNAscope-In Situ Hybridization (ISH)

RNAscope-ISH was carried out on sections obtained from FFPE tissue. Bond III Automated. ISH for HR-HPV E6/E7 mRNA was performed using the RNAscope 2.5 VS HPV HR18 probe (cat. #312599; Advanced Cell Diagnostics, Inc., Newark, CA, USA), which recognizes HPV 16, 18, 26, 31, 33, 35, 39, 45, 51, 52, 53, 56, 58, 59, 66, 68, 73 and 82, genotypes. ISH was performed on an automated platform (Discovery Ultra, Ventana, Roche, Basel, Switzerland) by using the RNAscope VS Universal HRP assay (Advanced Cell Diagnostics, Inc.), according to the manufacturer’s instructions. Positive staining has been identified as brown, punctate dots present in the nucleus and/or cytoplasm. Control probes for the bacterial gene 4-hydroxy-tetrahydrodipicolinate reductase (DapB), negative control, and for the housekeeping gene ubiquitin C (UbC), positive control for the evidence of preserved RNA, have also been included for each case.

### 4.4. Statistical Analysis

Because of the novelty of the study, whose primary objectives are mainly descriptive and exploratory, the minimum sample size has not been pre-determined.

Spearman correlation analysis and linear regression analysis were carried out using GraphPad Prism (version 5.0., GraphPad Software San Diego, CA, USA). A *p* < 0.05 was considered statistically significant.

## Figures and Tables

**Figure 1 ijms-24-03482-f001:**
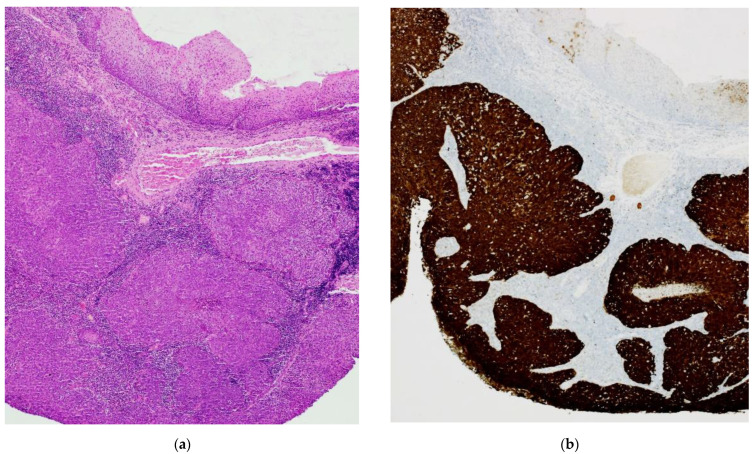
**HPV-OPSCC: morphology and immunohistochemistry** (**a**) H&E image of “basaloid” invasive squamous cell carcinoma; (**b**) Intense and diffuse positive IHC staining with anti-p16 antibody.

**Figure 2 ijms-24-03482-f002:**
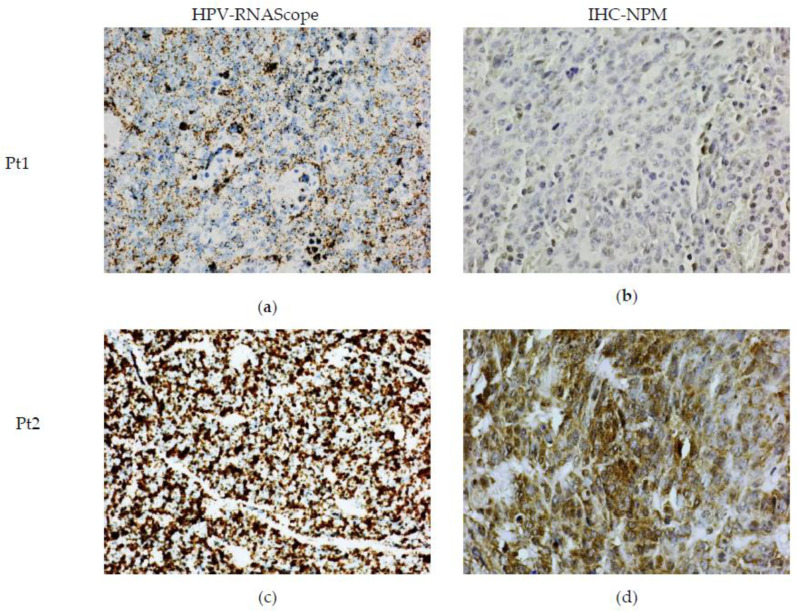
**HR-HPV ISH and IHC of NPM protein in OPSCC** (**a**,**c**) RNAScope ISH showing mild/moderate and large amounts of E6/E7 mRNA HR-HPV, respectively, (**b**,**d**) representative images of NPM IHC staining of same patients (Pt1, Pt2) depicted in (**a**,**c**). Weak and intense staining correlates with the amount of E6/E7 mRNA.

**Figure 3 ijms-24-03482-f003:**
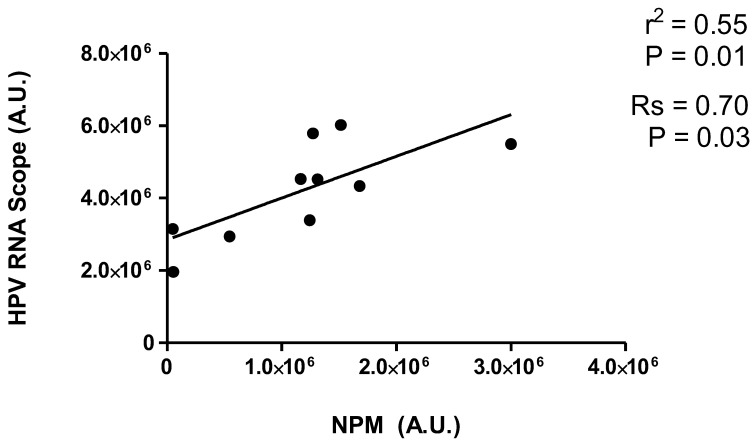
**Correlation analysis and linear regression between NPM levels and mRNA HR-HPV expression in OPCCS.** Spearman correlation analysis (Rs) and linear regression (r^2^) between NPM IHC positivity and mRNA HR-HPV ISH levels. IHC quantification of NPM and E6/E7 mRNA HR-HPV ISH were quantified by Image J color deconvolution of three adjacent fields for each section and transformed in arbitrary units (A.U.) of positivity for each patient.

## Data Availability

The datasets generated during and/or analyzed during the current study are available from the corresponding author upon reasonable request.

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
