# Peer review of "Positive Linear Relationship between Nucleophosmin Protein Expression and the Viral Load in HPV-Associated Oropharyngeal Squamous Cell Carcinoma: A Possible Tool for Stratification of Patients"

_ijms, 2023, doi:10.3390/ijms24043482_

Round 1

Reviewer 1 Report

Dear Authors, 

Its a novel idea to assess the Nucleophosmin Protein 2 Expression And The Viral Load In HPV-Associated Oropharyngeal Squamous Cell Carcinoma. the manuscript is well written . However few of the concerns are as follows . 

Abstract : Only ten patients were assessed and the authors have not observered any linear regression . Conclusion should be written with caution. Future direction could have beeen mentioned suggesting of more studies. 

Introduction : 

1.More details can be mentioned for NPM . Where it has been used and the rationale for using it in  HPV-Associated Oropharyngeal Squamous Cell Carcinoma . 

2. Research gap should be addressed . Which other component similar to NPM has been used previously and what is the need for the assessing NPM 

Material and methods : 

This should be the second main heading after the introduction . later results and discussion can be written . 

The information about the ethical approval from the institute is missing . kindly add . 

kindly add the region from where the tissues has been procured . 

Discussion : 

Kindly add studies which have been performed using these type of markers and correlate its finding with NPM .  

The discussion can be elaborated by including relevant studies . 

Regards 

Author Response

Reviewer 1

We thank the reviewer for acknowledging that “It’s a well written paper and it is novel idea to assess the Nucleophosmin Protein Expression And The Viral Load In HPV-Associated Oropharyngeal Squamous Cell Carcinoma”.

Q1: Abstract: Only ten patients were assessed, and the authors have not observed any linear regression.

A1. We thank the reviewer for this useful comment. We did not insert the statistical significance both in the figure (r2=0.55 P=0.01) and in the abstract of the regression, for a mistake. We now changed both the abstract and the figure, adding the P of the regression analysis. Although, we are aware that only ten patients were analysed, notwithstanding this, a significant positive correlation and a linear regression were observed.

Q2. Abstract: Conclusion should be written with caution. Future direction could have been mentioned suggesting of more studies. 

A2. We thank the reviewer for this comment. We added in the conclusion of the abstract “This study includes a small cohort of patients and cannot yield to conclusive statements. Further studies with large series of patients are needed to support this hypothesis”.

Q3 Introduction: 1. More details can be mentioned for NPM. Where it has been used and the rationale for using it in HPV-Associated Oropharyngeal Squamous Cell Carcinoma. 

A3: We added other papers to give the rationale for NPM analysis in HPV-Associated Oropharyngeal Squamous Cell Carcinoma in the Introduction. We described different papers that investigated the central role of the nucleolus and hence of nucleolar proteins in HPV infection. Indeed, several viruses, including HPV, have evolved molecular strategies to take control of nucleolar functions ultimately favouring viral replication and maturation, such as ribosomal biogenesis, cell cycle progression and apoptosis. In keeping with a nucleolar role of HPV infection, changes in numbers or in morphometric parameters of nucleoli, in particular of silver-stained nucleolar organizer regions (AgNORs) have been previously proposed as useful predictors of worse prognosis in different form of HPV infected squamous cell carcinomas. Nucleophosmin is a nucleolar protein implicated in different processes including ribosomal synthesis, cell cycle regulation, repair of DNA damage and centrosome duplication. Moreover, NPM is required in proliferation and inhibition of differentiation in primary keratinocyte cells expressing HPV E6 and E7, and also in the assembly of the HPV capsid. Starting from all these considerations we decided to assess the relationship between NPM expression and the HR-HPV viral load in tumour tissue to see whether a correlation is present. We added all this new part in the Introduction.

Q4: Introduction. Research gap should be addressed. Which other component similar to NPM has been used previously and what is the need for the assessing NPM.

A4: This paper, although in a small number of patients, could possibly open new useful possibilities in patients’ stratification and in the therapy decision. The of NPM immunohistochemical analysis together with HR-HPV RNAScope-ISH in routine practice could stratify patients with HPV-positive OPSCC into a group with relatively stable disease and a group with progression of malignancy. This stratification potentially could have clinical implications because NPM inhibition could offer therapeutic benefits to these patients. Moreover, since we demonstrated in previous papers that NPM can be considered an extracellular alarmin, the circulating expression of NPM could be also exploited in future applications as a predictor of HPV infection load by salivary or blood test screening. We added studies in support to our data, in which NPM inhibition was used in different viral infection including HPV positive cervical cancer, HIV infection and different solid tumours. Indeed, Phase I/II cancer clinical trials, using NPM inhibitor the CIGB-300 peptide was used with success in HPV positive cervical cancers and at the same time has been shown to be safe and well tolerated (and the synthetic pseudopeptide N6L (NCT01711398) in all solid tumours. CIGB-300 has been also tested in an HIV-infection model with success). Moreover 75% of the patients with cervical cancer, showed a reduction at colposcopy of tumour and 19% exhibited full histological regression. In addition, HPV DNA was negative in 48% of the previously positive patientsThe expression of different biomarkers has been linked to HPV carcinogenesis, given the importance of the nucleolus, different nucleolar proteins have been used to this aim.  For example, PITC1 and nucleolin are two nucleolar proteins that have been linked to uterine cervical cancer aggressivity, but also a different cellular protein expression i.e. proliferating cell nuclear antigen (PCNA), a protein essential for DNA replication. was associated with high-risk human papillomavirus (HPV) and the progression of cervical intraepithelial neoplasia (CIN), we added this comment in the Introduction and Discussion sections.

Q5: Material and methods: This should be the second main heading after the introduction. later results and discussion can be written. 

A5: Although we agree with the reviewer, we followed the IJMS word template, where the Materials and Methods are inserted after the Discussion section.

Q6: The information about the ethical approval from the institute is missing. kindly add. 

A6: We already inserted ethical committee of IDI-IRCCS hospital (Prot. n. 29 CE/2022 N° Registro Pareri: 698/1-2022) in the paragraph Institutional Review Board Statement.

Q7: kindly add the region from where the tissues have been procured. 

A7: We thank the reviewer for pinpointing this, we added, Rome, Italy to the hospital place. We had already written in Materials and Methods the origin of the tissue samples: “Ten cases of OPSCC were retrieved from archives of Histopathology Division of San Carlo di Nancy Hospital and Istituto Dermopatico dell’Immacolata (IDI-IRCCS), Rome-Italy”.

Q8. Discussion: Kindly add studies which have been performed using these types of markers and correlate its finding with NPM.  

A8: We thank the reviewer for this useful comment. We added relevant studies as indicated in A3 and A4

Q9: The discussion can be elaborated by including relevant studies. 

A9: We added in the Discussion other relevant studies as requested. See the answer A3 and A4. 

Reviewer 2 Report

The manuscript subject is interesting and important findings are shown regarding the expression of NPM in high-risk HPV-associated OPSCC. I would only suggest the authors to substitute "might be" to "could be" when they conclude the manuscript, both in "abstract" and "text" as follows: "...NPM IHC, together with HPV RNAScope, might be used as a predictor of transcriptionally active HPV presence and tumour progression." 

Despite the positive positive correlation between NPM expression and HR-HPV mRNA, more about this correlation needs to be investigated. 

Author Response

Reviewer 2

We thank the reviewer for acknowledging that “The manuscript subject is interesting and important findings are shown regarding the expression of NPM in high-risk HPV-associated OPSCC”.

Q1. I would only suggest the authors to substitute "might be" to "could be" when they conclude the manuscript, both in "abstract" and "text" as follows: "...NPM IHC, together with HPV RNAScope, might be used as a predictor of transcriptionally active HPV presence and tumour progression." 

A1 We followed reviewer’s suggestions and changed "might be" with could be, both in abstract and in the text.

Q2. Despite the positive correlation between NPM expression and HR-HPV mRNA, more about this correlation needs to be investigated. 

A2. We agree with the reviewer that more about this correlation needs to be investigated. The aim of this study was to add an additional biomarker to stratify severity (viral load) of HPV-infection besides RNAscope technique. Our findings, despite the small number of patients, show that the inhibition of NPM could have potential therapeutic implications since targeting NPM might impact simultaneously on viral replication and on the neoplastic state that emerges from it. We agree that absolutely, much more studies are needed. Since we demonstrated in previous papers that NPM can be considered an extracellular alarmin, the circulating expression of NPM could be also exploited in future applications as a predictor of HPV infection load by salivary or blood test screening. Therefore, this paper although in a small number of patients could possibly open new useful possibilities in patients’ stratification and in the therapy decision.

Indeed, Phase I/II cancer clinical trials, using NPM inhibitor the CIGB-300 peptide was used with success in HPV positive cervical cancers and at the same time has been shown to be safe and well tolerated and the synthetic pseudopeptide N6L (NCT01711398) in all solid tumours. CIGB-300 has been also tested in an HIV-infection model with success (. Moreover 75% of the patients with cervical cancer, showed a reduction at colposcopy of tumour and 19% exhibited full histological regression. In addition, HPV DNA was negative in 48% of the previously positive patients.

We added this comment in the Discussion section.

Reviewer 3 Report

This is a well-written and novel paper about a relationship between NMP leves and mRNA HR-HPV expression in OPCCS. However, additional experiments are recommended before accepting this manuscript.  Since the correlation coefficient is 0.7, a sample size of 10 would result in low power. Further increase in the number of samples is necessary to strengthen the conclusion, so please collect more samples and add more data.

Author Response

Reviewer 3

Q.1 This is a well-written and novel paper about a relationship between NMP levels and mRNA HR-HPV expression in OPCCS. However, additional experiments are recommended before accepting this manuscript.  Since the correlation coefficient is 0.7, a sample size of 10 would result in low power. Further increase in the number of samples is necessary to strengthen the conclusion, so please collect more samples and add more data.

A1. We agree with the reviewer that the sample size should be increased. In order to improve statistical power, the sample size should be at least doubled, but in few days, it will be impossible to accomplish it. Therefore, the results of the present study will justify future analyses on a higher number of samples, to confirm the results obtained. Since we demonstrated in previous papers that NPM can be considered an extracellular alarmin, the circulating expression of NPM could be also exploited in future applications as a predictor of HPV infection load by salivary or blood test screening. Therefore, this paper although in a small number of patients, could possibly open new useful possibilities in patients’ stratification with regard to the viral load and progression of disease. Our findings, despite the small number of patients, show that the inhibition of NPM could have potential therapeutic implications since targeting NPM might impact simultaneously on viral replication and on the neoplastic state that emerges from it.

Indeed, Phase I/II cancer clinical trials, using NPM inhibitor the CIGB-300 peptide was used with success in HPV positive cervical cancers and at the same time has been shown to be safe and well tolerated and the synthetic pseudopeptide N6L (NCT01711398) in all solid tumours. CIGB-300 has been also tested in an HIV-infection model with success. Moreover 75% of the patients with cervical cancer, showed a reduction at colposcopy of tumour and 19% exhibited full histological regression. In addition, HPV DNA was negative in 48% of the previously positive patients. Thus, NPM inhibition could be considered a possible therapeutic intervention also in HPV-positive OPSCC. We added this comment in the Discussion section.

Round 2

Reviewer 3 Report

Although the validity of the conclusions is questionable due to the small number of specimens, the author's conclusions are more convincing with the addition of further perspectives for the future and more references to guide their conclusions. I consider the paper sufficient to be accepted.